# Design principles for NASICON super-ionic conductors

Jingyang Wang[1,2,3], Tanjin He [1,2], Xiaochen Yang[1,2], Zijian Cai [1,2], Yan Wang [4], Valentina Lacivita[4], Haegyeom Kim[1], Bin Ouyang[5] ✉ & Gerbrand Ceder [1,2] ✉

Na Super Ionic Conductor (NASICON) materials are an important class of solid-state electrolytes owing to their high ionic conductivity and superior chemical and electrochemical stability. In this paper, we combine first-principles calculations, experimental synthesis and testing, and natural language-driven text-mined historical data on NASICON ionic conductivity to achieve clear insights into how chemical composition influences the Na-ion conductivity. These insights, together with a high-throughput first-principles analysis of the compositional space over which NASICONs are expected to be stable, lead to the successful synthesis and electrochemical investigation of several new NASICONs solid-state conductors. Among these, a high ionic conductivity of 1.2 mS cm$^{-1}$ could be achieved at 25 °C. We find that the ionic conductivity increases with average metal size up to a certain value and that the substitution of $PO_4$ polyanions by $SiO_4$ also enhances the ionic conductivity. While optimal ionic conductivity is found near a Na content of 3 per formula unit, the exact optimum depends on other compositional variables. Surprisingly, the Na content enhances the ionic conductivity mostly through its effect on the activation barrier, rather than through the carrier concentration. These deconvoluted design criteria may provide guidelines for the design of optimized NASICON conductors.

Solid-state batteries are a promising next-generation battery technology because of their potential for improvements in safety and energy density, stemming from the replacement of the conventional flammable organic electrolyte by a dense layer of a solid-state conductor that serves as both the alkali-ion-conducting electrolyte and separator[1,2]. Numerous efforts have thus been devoted to the discovery of solid-state conductors with high ionic conductivity as well as good chemical and electrochemical stability[3–9]. Among these, sodium superionic conductors, or NASICONs for short, are of considerable interest. NASICON materials are a class of polyanionic materials with the general formula $Na_xM_2(AO_4)_3$, where M represents a transition or main group metal and $AO_4$ represents a

polyanion[10,11]. A robust crystalline framework with the rhombohedral (R-3c) symmetry is constructed by corner-sharing metal octahedra and polyanion tetrahedra, as illustrated in Fig. 1a, which creates two distinct Na sites. Therefore, Na ions can migrate from one site to another through two triangular bottlenecks and form a three-dimensional percolating channel for fast ion conduction[12]. NASICON compounds can also form in a monoclinic symmetry (C2/c) depending on the composition and temperature, which splits the Na2 site into two different sites[12]. However, the framework skeleton remains the same, and thus, the ion-transport channel is similar. The high ionic conductivity and structural stability of NASICON-type materials are of great interest for building solid-state batteries and

[1]Materials Sciences Division, Lawrence Berkeley National Laboratory, Berkeley, CA 94720, USA. [2]Department of Materials Science and Engineering, University of California, Berkeley, CA 94720, USA. [3]School of Sustainable Energy and Resources, School of Materials Science and Intelligent Engineering, Nanjing University, Suzhou, China. [4]Advanced Materials Lab, Samsung Advanced Institute of Technology and Samsung Semiconductor, Inc, Cambridge, MA 02138, USA. [5]Department of Chemistry and Biochemistry, Florida State University, Tallahassee, FL 32306, USA. ✉e-mail: bouyang@fsu.edu; gceder@berkeley.edu

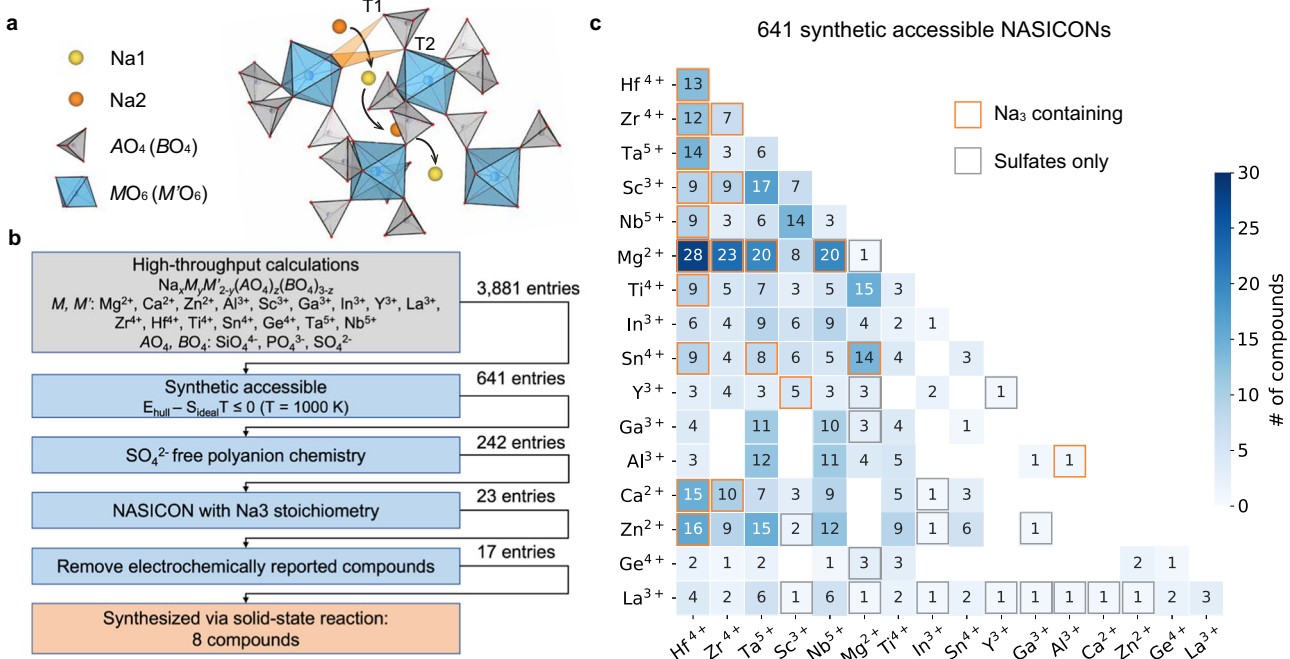

**Fig. 1 | Heat map of synthetically accessible NASICONs predicted by DFT.**
**a** Crystal structure of a rhombohedral NASICON showing the Na-ion conduction pathway. Two bottleneck triangles are denoted as T1 and T2. **b** High-throughput computation workflow. **c** Distribution of the 641 DFT calculated synthetically accessible NASICONs. The color map indicates the number of stable compounds containing a specific metal pair. The orange squares indicate blocks including NASICONs with Na₃ stoichiometry, and the grey squares mark blocks containing only sulfate-type NASICONs.

other applications such as ion-selective membranes and gas-sensing devices[13–18].

The history of NASICON-type materials can be traced back to the 1960s. Hong and Goodenough et al. reported the prototype of NASICON, $Na_3Zr_2(SiO_4)_2(PO_4)$. They discovered that the polyanion composition is quite flexible as the ratio between $SiO_4$ and $PO_4$ can be finely tuned, forming a complete solid-solution $Na_{1+x}Zr_2(SiO_4)_x(PO_4)_{3-x}$ ($0 \leq x \leq 3$), with the Na content varying from 0 to 4 as required by charge neutrality[10,11]. The ionic conductivity of this solid solution at 300 °C achieved an optimal value of 0.2 S cm⁻¹ when $x = 2$[11]. However, the specific roles of the polyanion composition and Na content in determining the ionic conductivity have not been strictly decoupled[19]. Different polyanion groups, such as $AsO_4^{3-}$, $SO_4^{2-}$, and $SeO_4^{2-}$, can be incorporated in the NASICON structure[20,21], though compounds with these polyanions have not been studied as extensively as the Si–P system. The ability of the NASICON structure to accommodate different ions on the M site has been extensively demonstrated. Divalent ($Mg^{2+}$, $Ca^{2+}$, etc.), trivalent ($Al^{3+}$, $Ga^{3+}$, $Sc^{3+}$, etc.), tetravalent ($Zr^{4+}$, $Hf^{4+}$, $Ti^{4+}$, etc.), and pentavalent ($Nb^{5+}$, $Sb^{5+}$, etc.) metal cations can all be stabilized in the M site with relatively large solubilities[22,23], and the ionic conductivity can be increased by one order of magnitude using aliovalent doping strategies[14,23]. However, because these compositional variables (e.g., Na content, metal and polyanion species) are highly convoluted, their distinct effect on the ionic conductivity is challenging to differentiate and rationalize. For example, replacing $Zr^{4+}$ with lower-valent $Mg^{2+}$ also increases the Na content[24], Moreover, because the NASICON framework can accommodate a wide range of chemical species in the M and A sites[11,21,23,25–35], the possible compositional space is immense[36]. Although we systematically studied the compositional bounds on phase stability of NASICONs in recent work[36], a general picture of the relationship between the compositional variables of NASICON materials and their conductivities, is essential to provide guiding principles for the design of high-performance NASICON conductors in such a large, yet partially uncharted chemical space.

In this study, we combine ab-initio calculations with text-mined data on the ionic conductivity of NASICONs from a corpus of several million papers and our experimental work to disentangle the factors that control the Na-ion conductivity in NASICONs. High-throughput density-functional-theory (DFT) calculations are first used to investigate the stability of NASICONs. A subset of the predicted compositions is experimentally explored, leading to the successful synthesis of eight NASICONs. Five of these were introduced in our recent work[35] and three are new. In this paper we study the electrochemical performance of these eight recently discovered compounds as solid-state conductors, and present detailed crystal structure and electrochemical impedance. The ionic conductivity and activation energy of the as-synthesized NASICONs were investigated, and high ionic conductivity (~10⁻⁴ S cm⁻¹) at room temperature (~25 °C) was found in $Na_3HfZr(SiO_4)_2(PO_4)$ and $Na_3HfSc(SiO_4)(PO_4)_2$. Natural language processing (NLP) tools were applied to text-mine NASICON literature and identify the trends of the ionic conductivity evolution with respect to different compositional variables. Guided by this text-mined information, we improved the NASICON room-temperature (~25 °C) ionic conductivity in this work up to 1.2 mS cm⁻¹, which is one of the highest conductivities among reported NASICONs. Ab-initio molecular dynamic (AIMD) simulations with well-controlled compositional variables revealed the following comprehensive compositional optimization strategies for NASICON conductors: (1) the optimal Na content is ~$x = 3$ but depends on the polyanion chemistry, (2) high ionic conductivity is found in compositions with a large cation size until the size reaches an optimal value, and (3) high silicate content enhances the ionic conductivity. Our work provides a comprehensive guiding map to facilitate the discovery of new NASICONs and insights into the optimization of NASICON ionic conductivity toward the application of solid-state Na batteries.

## Results

### Map of synthetically accessible NASICONs predicted by high-throughput DFT calculations

High-throughput ground-state calculations were conducted to evaluate the phase stability of the NASICON structure across a

wide range of compositions. The screening criteria are summarized in Fig. 1b. We considered the general formula $Na_xM_yM'_{2-y}$ $(AO_4)_z(BO_4)_{3-z}$, i.e., both the metal site and polyanion site can be occupied by up to two species. In particular, we considered $SiO_4^{4-}$, $PO_4^{3-}$ and $SO_4^{2-}$ as possible polyanions and 16 possible electrochemical inactive metals for $M$ and $M'$. (Fig. 1b). As the Na content $x$ can take values from 0 to 4.0, $y$ can take values from 0 to 2.0, and $z$ can take values from 0 to 3.0. Using intervals of 0.5 for sampling these composition variables, 3881 charge-balanced compositions were enumerated, and their energy above the convex hull ($E_{hull}$) in the relevant chemical space was calculated using DFT. To guide the experimental exploration at high temperature[37], we further computed the ideal configurational entropy ($S_{ideal}$, see Methods) of each composition. We obtained 641 NASICONs with $E_{hull} - S_{ideal}T \leq 0$ ($T = 1000$ K), making them likely to be synthetically accessible. To narrow the target to possible solid-state conductors, $SO_4^{2-}$-containing NASICONs were excluded at this point because they are generally less stable at high temperature and therefore challenging to densify[28]. Of the 242 remaining $SO_4^{2-}$-free NASICONs, only 23 compounds with $Na_3$ stoichiometry were further considered, within which the electrochemical performance of 17 compounds has not yet been reported (Supplementary Table 1). Finally, 8 of these NASICON compounds were successfully synthesized through a classic solid-state synthesis. The distribution of the 641 NASICONs predicted to be synthetically accessible is presented in Fig. 1c, sorted by their metal chemistry. The color intensity and number in each box indicate the number of stable compounds found that contain both metal elements. The heat map is further annotated as follows: (1) blocks that contain at least one $Na_3$ compound are highlighted with orange rectangles; (2) blocks that only have sulfate NASICONs, and therefore not included in the experimental study, are marked with grey rectangles.

## Experimental exploration of the predicted NASICONs as potential solid conductors

Experimental exploration via solid-state methods of the 17 NASICONs mentioned above led to the synthesis of 8 NASICON compounds. Supplementary Table 1 summarizes all the synthesis attempts, including failures. Although many silicate NASICONs containing metals with 5+ oxidation state were predicted to be synthesizable, they tended to form $NaM^{5+}O_3$ competing phases. Some failed synthesis attempts also involved the melting of precursors, even at relatively low temperatures (~600 °C). Note that for all the synthesis attempts, only the calcination temperatures were varied (600–1100 °C) to optimize the synthesis outcome; therefore, other variables such as the precursor selection, mixing, calcination atmosphere, or even alternative synthesis routes might be considered to access those phases[38–40].

X-ray diffraction (XRD) patterns of the successfully synthesized NASICONs are presented in Fig. 2. Among the 8 compositions, only $Na_3HfZrSi_2PO_{12}$ has a monoclinic structure (C2/c), whereas all the others crystallize in a rhombohedral (R-3c) structure. Representative Rietveld refinement profiles for the monoclinic ($Na_3HfZrSi_2PO_{12}$) and rhombohedral ($Na_3HfScSiP_2O_{12}$) structures are presented in Figs. 2a and 2b, respectively, both of which show a small amount of impurity phase, consistent with previous findings in the literature[41]. Supplementary Fig. 1 presents the refined XRD patterns for other NASICONs, with the extracted lattice parameters summarized in Supplementary Fig. 2, and other refined structural parameters summarized in Supplementary Table 2–9. In general, the $c$ lattice parameter is mainly affected by the polyanion species, which determine the unit-cell volume, with a larger volume for increasing silicate content. The $a$ lattice parameter, instead, is primarily controlled by the metal species.

Dense pellets of the as-synthesized materials were prepared (Methods and Supplementary Table 10) for the electrochemical impedance spectroscopy (EIS) tests. The bulk and grain-boundary

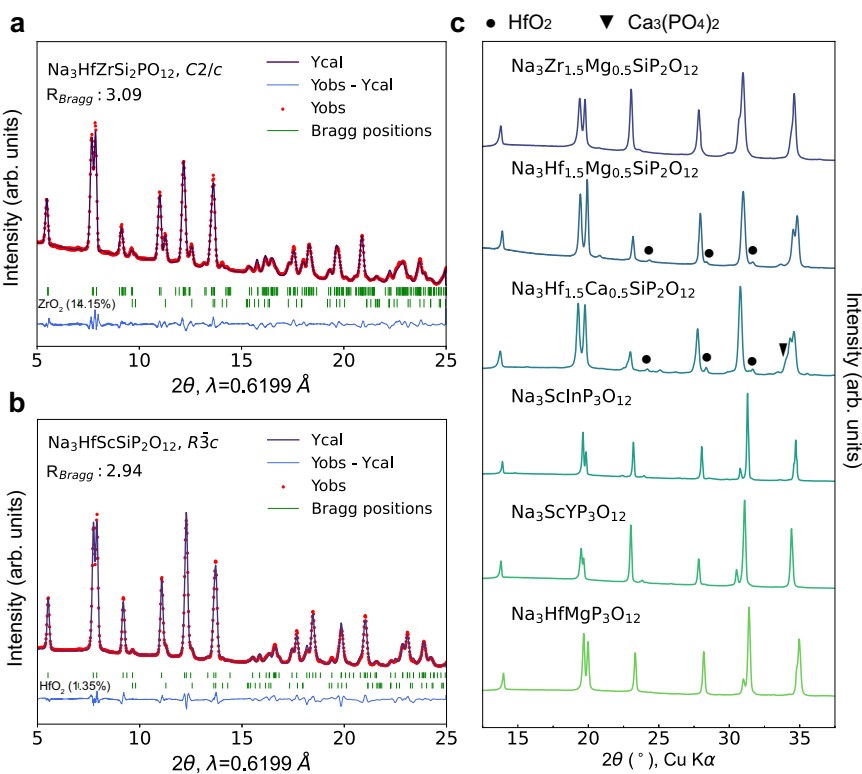

**Fig. 2 | XRD patterns of the as-synthesized NASICONs.** Rietveld refinement results of synchrotron XRD data of $Na_3HfZrSi_2PO_{12}$ with monoclinic C2/c symmetry (**a**), and of $Na_3HfScSiP_2O_{12}$ with rhombohedral R-3c symmetry (**b**). **c** Lab XRD patterns of other as-synthesized NASICONs. The corresponding refinement results are presented in Supplementary Fig. 1, and structural parameters in Supplementary Table 2–9.

conductivities at room temperature (~25 °C) are summarized in the top panel of Fig. 3, and the corresponding Nyquist plots, equivalent circuits and fitting parameters are presented in Supplementary Fig. 3–4, Supplementary Table 11, respectively. In Fig. 3, the compounds are grouped by their polyanion chemistry, i.e., phosphates $(PO_4)_3$ and mixed silicate-phosphates $(SiO_4)(PO_4)_2$ and $(SiO_4)_2(PO_4)$. In each polyanion group, compounds are sorted by ascending average metal radius $(\bar{r}_M)$. In general, the ionic conductivity increases as the silicate content increases[19], irrespective of the cations. Specifically, the total conductivities of the pure phosphate group are in the range of $10^{-6}$ to $10^{-5}$ S cm$^{-1}$, whereas those of the $(SiO_4)(PO_4)_2$ group are in the range of $10^{-5}$ to $10^{-4}$ S cm$^{-1}$. The highest ionic conductivity of $4.4 \times 10^{-4}$ S cm$^{-1}$ is achieved in $Na_3HfZr(SiO_4)_2(PO_4)$. Within each polyanion group, the ionic conductivity shows a non-monotonic trend with $\bar{r}_M$. Indeed, the ionic conductivity increases with $\bar{r}_M$ only up to an optimal value. For example, the ionic conductivity increases from $2.40 \times 10^{-6}$ to $6.48 \times 10^{-5}$ S cm$^{-1}$ in going from $Na_3HfMg(PO_4)_3$ ($\bar{r}_M = 0.715$ Å) to $Na_3ScIn(PO_4)_3$ ($\bar{r}_M = 0.773$ Å) and from $6.17 \times 10^{-5}$ S cm$^{-1}$ for $Na_3Hf_{1.5}Mg_{0.5}(SiO_4)(PO_4)_2$ ($\bar{r}_M = 0.713$ Å) to $1.87 \times 10^{-4}$ S cm$^{-1}$ for $Na_3HfSc(SiO_4)(PO_4)_2$ ($\bar{r}_M = 0.728$ Å). Further increasing $\bar{r}_M$ is unfavorable with both polyanion compositions, as $Na_3ScY(PO_4)_3$ ($\bar{r}_M = 0.823$ Å) and $Na_3Hf_{1.5}Ca_{0.5}(SiO_4)(PO_4)_2$ ($\bar{r}_M = 0.783$ Å) appear to have lower conductivities of $4.10 \times 10^{-6}$ and $2.93 \times 10^{-5}$ S cm$^{-1}$, respectively. The refined compositions are listed in Supplementary Table 2 – 9, and agree reasonably with the nominal composition, so that the trend of the average cation radius presented in Fig. 3 remains valid. Therefore, nominal compositions are used in the analysis for simplicity. However, Na content <3 are observed for the refined compositions, probably due to the Na loss during the high temperature annealing.

The ionic conductivity of each NASICON was measured at various temperatures (Supplementary Fig. 6), revealing typical Arrhenius behavior, and the extracted activation energies are plotted in the bottom panel of Fig. 3. Consistent with the trend in the conductivities, as the silicate content increases, the activation energy generally decreases. Similarly, the activation energy is reduced when large M

cations are present, though there is a limit to the benefits of this size effect. $Na_3HfZr(SiO_4)_2(PO_4)$, which has the highest ionic conductivity, also exhibits the lowest bulk activation energy of 0.302 eV. These experimental observations indicate that for our as-synthesized NASICONs, the ionic conductivity is primarily determined by the activation energy. It is worth noting that though the nominal Na content of the as-synthesized compounds are all equal to 3, the ratio between Na1, Na2 site occupancies can be different (Supplementary Table 2–9, Supplementary Fig. 7). The Na1 site occupancy decreases as the silicate content increases[19]: the three NASICONs with pure phosphates have Na1 occupancy of ~0.8, whereas the monoclinic $Na_3HfZr(SiO_4)_2(PO_4)$ phase with the highest ionic conductivity exhibits the lowest Na1 occupancy of ~0.63.

## Further compositional optimization based on text-mined ionic conductivity data

To generalize our findings, we investigated a text-mined dataset containing the experimentally measured ionic conductivity of 475 reported NASICONs. The corresponding papers were identified by screening over two million materials science articles with criteria for specific materials (i.e., NASICONs) and properties (i.e., ionic conductivity) via chemical named entity recognition[42–44]. Three compositional variables were used to visualize the distribution of the ionic conductivity data obtained from the literature: the Na content $(x)$, average metal radius $(\bar{r}_M)$, and average A site radius $(\bar{r}_A)$. As the ionic radius of $Si^{4+}$ (0.26 Å) is larger than that of $P^{5+}$ (0.17 Å), the variation of the silicate content in Fig. 3 can be appropriately reflected by $\bar{r}_A$. Therefore, a three-dimensional compositional space of the NASICONs was constructed, as shown in Fig. 4a, with the color of each point indicating its ionic conductivity on a log scale. Overall, the ionic conductivity of NASICONs varies greatly with composition, ranging from $10^{-14}$ to $10^{-3}$ S cm$^{-1}$. However, this finding should be taken with caution because some of this difference may also result from inconsistent sample preparation or testing methods between the various literature studies. In particular, pellet density and varying interfacial resistance may affect the measured ionic conductivity. These experimental details are often not explicitly reported in the literature[19]. In Fig. 4a, most high ionic conductivity values can be found in certain narrow subspaces. For instance, two thin sections with $0.70 \le \bar{r}_M \le 0.74$ Å and $2.5 \le x \le 3.5$ are highlighted, respectively, and the projections of data points within those two sections are plotted in Fig. 4b and Fig. 4c.

Figure 4b indicates that, within the group of pure phosphates ($\bar{r}_A = 0.17$ Å), the high-ionic-conductivity compositions have Na content around $x = 2.5$ (though the limited number of Na-rich phosphates that have been reported limit the confidence level of this finding). As the silicate content increases ($\bar{r}_A$ increasing), the Na content, at which point the ionic conductivity is maximized shifts toward higher values. For NASICONs with silicate contents equal to or larger than two (per formula unit of three polyanion groups), the optimal Na content is between 3.0 and 3.4. Note that the optimal Na content has already been investigated in previous studies[19,23,36,45]. However, we argue that the optimal value might be different for different polyanion compositions, and in the following section we investigate the Na content effect in more depth by setting other compositional variables constant in AIMD simulations. Figure 4c shows that the maximum ionic conductivity occurs when $\bar{r}_M$ is slightly above 0.72 Å (though the exact optimal value might be dependent on polyanion species), whereas NASICONs with a $\bar{r}_M$ that is too small or too large are less likely to exhibit high ionic conductivity. Some exceptions are found among phosphate NASICONs with large lanthanide elements such as Er or Dy as doping cations that exhibit high ionic conductivity[46], however, the temperature at which EIS measurements were taken was not explicitly mentioned.

Among all the NASICONs synthesized in our limited set of experiments, $Na_3HfZrSi_2PO_{12}$ exhibits the highest ionic conductivity.

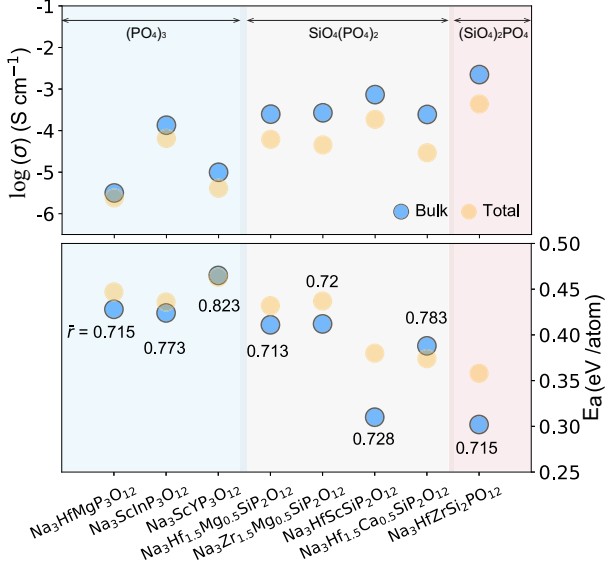

**Fig. 3 | Measured ionic conductivity of as-synthesized NASICONs.** Scatter plots of room-temperature (~25 °C) conductivities (log scale, upper panel) fitted from Nyquist plots (Supplementary Fig. 3) and activation energies (lower panel) extracted from the linear fitting of the Arrhenius plots (Supplementary Fig. 6). The bulk properties are indicated by blue circles, whereas the total properties are indicated by orange ones. On the x-axis, the compounds are grouped by polyanion chemistry, and in each group, compounds are ordered by increasing average metal radius ($\bar{r}_M$, in Å). $\bar{r}_M$ of each compound is also labeled in the bottom panel.

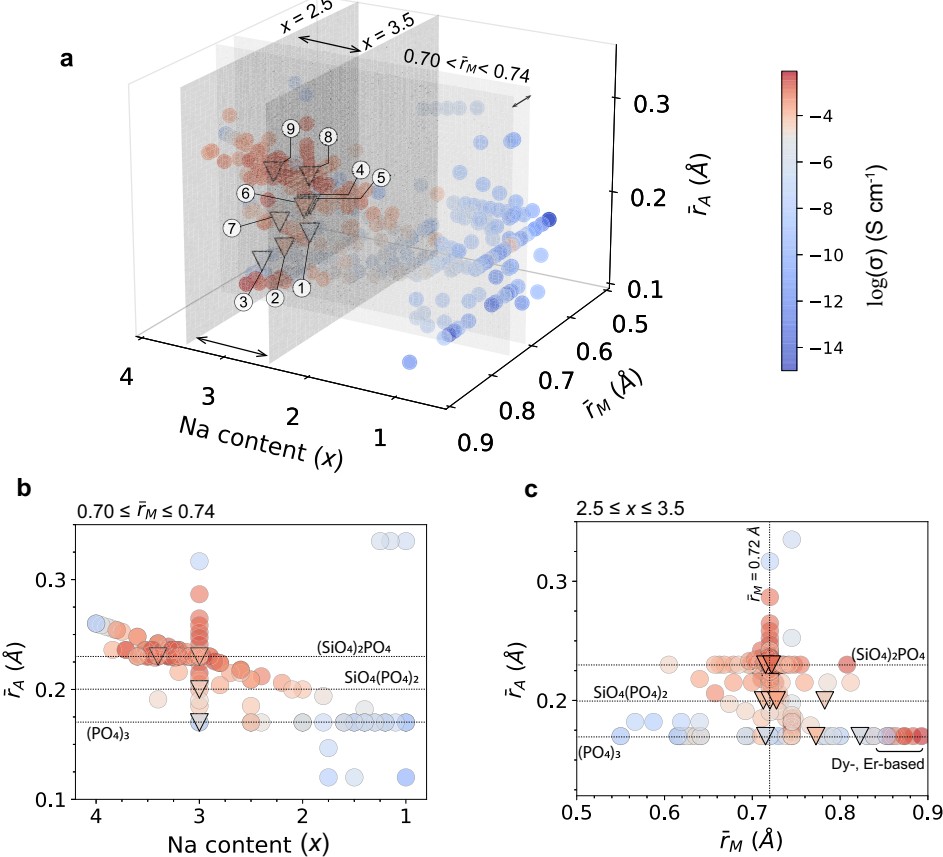

**Fig. 4 | Reported room-temperature (~ 25 °C) NASICON conductivities as a function of compositional variables. a** Three-dimensional scatter plot of the reported NASICON conductivities as a function of Na content ($x$), average M site radius ($\bar{r}_M$), and average A site radius ($\bar{r}_A$). The ionic conductivity (on a log scale) of each NASICON is represented by the color bar. **b** Projection of data points between $\bar{r}_M = 0.70$ and $\bar{r}_M = 0.74$ Å in **a** on the Na content $\times \bar{r}_A$ plane. **c** Projection of the data points between $x = 2.5$ and $x = 3.5$ in **a** on the $\bar{r}_M \times \bar{r}_A$ plane. The as-synthesized NASICONs in this work are represented by triangles: 1: $Na_3HfMg(PO_4)_3$, 2: $Na_3ScIn(PO_4)_3$, 3: $Na_3ScY(PO_4)_3$, 4: $Na_3Hf_{1.5}Mg_{0.5}(SiO_4)(PO_4)_2$, 5: $Na_3Zr_{1.5}Mg_{0.5}(SiO_4)(PO_4)_2$, 6: $Na_3HfSc(SiO_4)(PO_4)_2$, 7: $Na_3Hf_{1.5}Ca_{0.5}(SiO_4)(PO_4)_2$, 8: $Na_3HfZr(SiO_4)_2(PO_4)$, 9: $Na_{3.4}Hf_{0.6}Sc_{0.4}ZrSi_2PO_{12}$.

However, the data in Fig. 4 indicate that the ionic conductivity might still be further improved by modifying the average cation radius and stoichiometry of $Na_3HfZrSi_2PO_{12}$. Therefore, we attempted to use $Sc^{3+}$ as a large-size dopant with a lower valence to bring both the average cation radius and Na content toward the optimized values. By replacing 0.2, 0.4 $Hf^{4+}$ per *f.u.* with $Sc^{3+}$, the average cation radius can be increased from 0.715 Å to 0.719 and 0.722 Å, and the Na content to 3.2 and 3.4, respectively. Figure 5a presents the XRD patterns of Sc-substituted $Na_{3.2}Hf_{0.8}Sc_{0.2}ZrSi_2PO_{12}$ and $Na_{3.4}Hf_{0.6}Sc_{0.4}ZrSi_2PO_{12}$. Both compounds show a majority of NASICON phase with a small amount of $ZrO_2$ impurity. As the Sc content increases, $Na_{3.4}Hf_{0.6}Sc_{0.4}ZrSi_2PO_{12}$ no longer retains the monoclinic (*C2/c*) symmetry and can be properly refined with a rhombohedral model (Supplementary Fig. 5). EIS spectra of both compounds using Na metal as electrodes are presented in Fig. 5b. Compared with pristine $Na_3HfZrSi_2PO_{12}$, both $Na_{3.2}Hf_{0.8}Sc_{0.2}ZrSi_2PO_{12}$ and $Na_{3.4}Hf_{0.6}Sc_{0.4}ZrSi_2PO_{12}$ exhibit higher total ionic conductivity of 0.48 and 1.2 mS cm$^{-1}$, respectively. In particular, 1.2 mS cm$^{-1}$ is one of the highest conductivities among reported NASICON conductors at room temperature (~ 25 °C)[47,48]. It is worth noting that the ionic conductivity of this material may still be improved by increasing the silicate content, as inferred from the experimental and text-mined data, or by optimizing the sintering condition via hot pressing or spark plasma sintering techniques, instead of the cold-pressing method used in this work. However, the former essentially requires the incorporation of high-valence charge compensators, which may be inhibited by the challenges related to the synthesis of

materials with 5+ metals and high silicate content (Supplementary Table 1).

The stability of $Na_{3.4}Hf_{0.6}Sc_{0.4}ZrSi_2PO_{12}$ against a Na-metal anode was evaluated by constructing a Na | $Na_{3.4}Hf_{0.6}Sc_{0.4}ZrSi_2PO_{12}$ | Na symmetric cell. As shown in Fig. 5c, stable Na stripping and plating are achieved, and the small overpotentials are stable throughout the 200-h test. EIS spectra of the Na-metal symmetric cell were collected after 100-h and 200-h cycling, showing no obvious change in the overall impedance (Supplementary Fig. 9), indicating that the NASICON–Na metal interface is stable.

## Discussion

Our experimental analysis, combined with the large-scale extraction of ionic conductivity data from the literature, establishes optimal ranges for Na content, polyanion chemistry, and average metal radius. To better understand how these various compositional variables affect the ionic conductivity, ab initio molecular dynamic (AIMD) simulations were performed, and the results are presented in Fig. 6. Three different NASICON groups with different controlled variations in composition were designed so as to separate the effect of each variable as independently as possible:

1. Na-content variation: The polyanion chemistry per formula unit was fixed to be either $(PO_4)_3$ or $(PO_4)_2(SiO_4)$ with the average cation radius kept near 0.715 Å while varying the Na content by modifying the ratio between $Hf^{4+}$ (0.71 Å) and $Mg^{2+}$ (0.72 Å) with similar size.

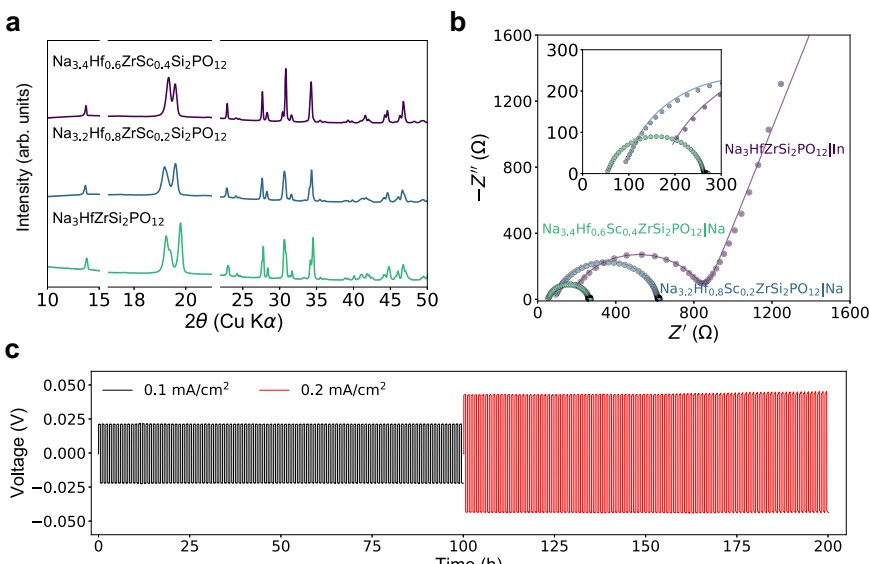

**Fig. 5 | Optimization of ionic conductivity via Sc doping. a** XRD patterns. The peaks between 18° and 20° are highlighted to show the change in crystal symmetry. **b** Nyquist plots obtained from EIS at room temperature (~25 °C). Indium metal was used as blocking electrodes for $Na_3HfZrSi_2PO_{12}$, whereas Na metal was used as the electrodes for $Na_{3.2}Hf_{0.8}Sc_{0.2}ZrSi_2PO_{12}$ and $Na_{3.4}Hf_{0.6}Sc_{0.4}ZrSi_2PO_{12}$. **c** Na stripping and plating tests in a temperature chamber at 25 °C at various current rates of a Na| $Na_{3.4}Hf_{0.6}Sc_{0.4}ZrSi_2PO_{12}$ | Na symmetric cell.

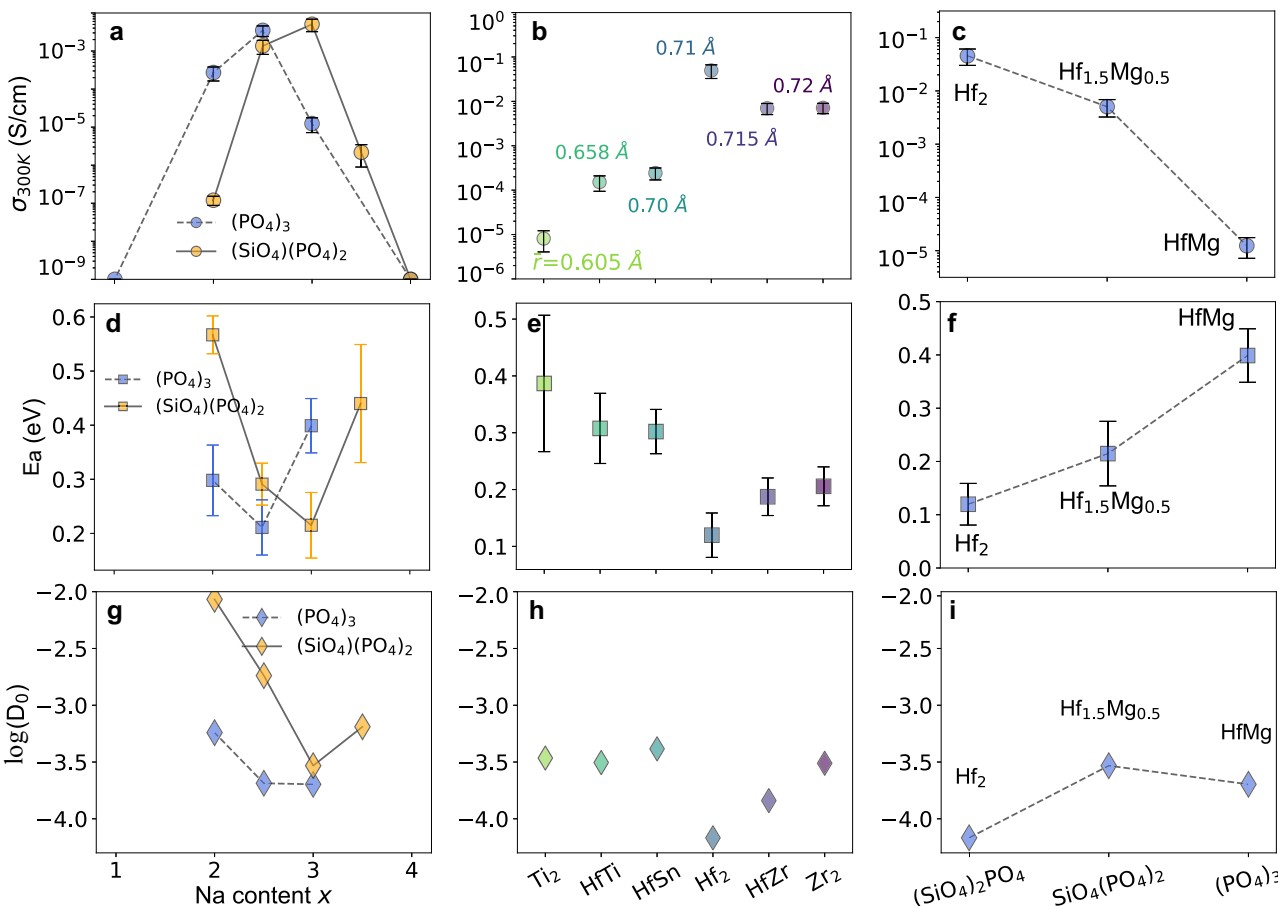

**Fig. 6 | Bulk ionic conductivities, activation energies, and prefactors calculated by AIMD.** Calculated room-temperature (300 K) bulk ionic conductivities (**a**) and activation energies (**d**) with error bars[77], and prefactors (**g**) of $Na_{1+2y}Hf_{2-y}Mg_yP_3O_{12}$ (y = 0, 0.5, 0.75, 1, 1.5, labeled blue) and $Na_{2+2z}Hf_{2-z}Mg_zSiP_2O_{12}$ (z = 0, 0.25, 0.5, 0.75, 1, labeled yellow). In these two groups, both the polyanion composition and cation radius are kept constant. Note that for compounds with y = 0, 1.5 and z = 1, the conductivities were not converged and therefore set as « $10^{-9}$ mS cm$^{-1}$. **b, e, h** Group of $Na_3MM'Si_2PO_{12}$ (M, M' = Ti, Hf, Sn, Zr), with constant Na content x and polyanion composition. **c, f, i** Group of $Na_3Hf_{2-z}Mg_zSi_{2-2z}P_{1+2z}O_{12}$ (z = 0, 0.5, 1) where $\bar{r}_M$ and the Na content x are kept constant. The error bars quantifying the statistical confidence levels of diffusion results from AIMD are calculated according to He et al. [77].

2.  Variation of the cation size: The Na content was fixed to be $Na_3$ per formula unit with $(PO_4)(SiO_4)_2$ as the polyanion while varying the average cation radius by modifying the combination of different $M^{4+}$ metal cations.

3.  Polyanion composition: The Na content was fixed to be $Na_3$ per formula unit with the cation radius kept at ~0.715 Å while modifying the polyanion and charge compensating through the ratio between $Hf^{4+}$ (0.71 Å) and $Mg^{2+}$ (0.72 Å).

Figure 6 shows the AIMD calculated ionic conductivity at 300 K (Fig. 6a–c), the activation energy (Fig. 6d–f), and the prefactor for Na diffusion (Fig. 6g–i) for the three compound groups. The correlations between the ionic conductivity and compositional variables in Fig. 6a–c are generally consistent with our experimental and text-mined ionic conductivity trends. Regarding the Na content effect, in Figs. 6a and 6d, conductivities and activation energies vary simultaneously with Na content, indicating that Na content not only determines the charge-carrier concentration, as expected, but also affects the ionic conductivity through the migration barrier. The correlation between Na content and migration barrier can be rationalized as follow: the migration barrier is determined by the size of the bottleneck for ion hopping, which is then controlled by the lattice parameter. For NASICONs with 1Na per formula unit, Na ion sits in the Na1 (6b) site that is face-sharing with two $MO_6$, as shown in Fig. 1a. Increasing Na content will displace the $Na^+$ into the Na2 (18e) site, which results in reduced occupancy of the Na1 site and increased electrostatic repulsion between $MO_6$ units, leading to larger unit cell volume and therefore bottleneck size[37]. This trend is supported by the correlation between Na content and lattice parameter, as shown in Supplementary Fig. 10. Furthermore, consistent with the text-mined data in Fig. 4b, Fig. 6a also shows that the optimal Na content for ionic conductivity is polyanion dependent: for phosphates the optimal value is 2.5 while for mixed polyanions the optimal value is near 3. Correspondingly in Fig. 6d, the Na content that is optimal for lowering the migration barrier is also lower for phosphates than for mixed polyanions. This result indicates that the optimal Na content for different polyanion composition appears to be mostly driven by Na's effect on the activation energy rather than by its direct relation to the carrier concentration.

The correlation between the cation radius and activation energy, displayed in Fig. 6e, is consistent with our experimental observation, i.e., the activation energy generally decreases with increasing cation radius but becomes mostly insensitive once surpassing the optimal value. Our finding is also in line with previous studies indicating that an optimal cation radius exists[23,47], and here we demonstrate this effect more rigorously by keeping other compositional variables constant. The results in Fig. 6f show that by keeping the Na content and cation radius constant, the activation energy can be reduced up to ~200 meV solely by increasing the silicate content from 0 to 2/3, implying that the highest ionic conductivity might be obtained in pure silicate NASICONs or NASICONs with $SiO_4^{4-}$ content above 2/3. However, the experimental validation of this optimum is challenging as synthesis attempts for this type of NASICON were unsuccessful via the solid-state method (Supplementary Table 1). Increasing the Na content, metal radius, and/or Si content all lead to the increase of the lattice constant of NASICON and thereby enhance the ionic conductivity. This observation would indicate that Na mobility is mostly controlled by the size of some migration bottleneck, and not by other effect such as cation-cation interaction[49–52], topology[5,51,53,54], or polyanion-assisted motion[55–60].

While the dominant effect of the compositional variables is likely the control of the size of the ion migration bottleneck and, therefore, the activation energy, the composition also has a more subtle effect on the ordering tendency of Na ions and lattice softness of the NASICON framework, which is reflected in the variation of the prefactor for Na diffusion. The evolution of the prefactor differs from that of the activation energy and ionic conductivity. Generally, it becomes smaller as the ionic conductivity becomes larger, consistent with prior findings in the literature[61–63]. When tuning Na content and metal radius (Figs. 6g, 6h), the prefactor is minimal when the ionic conductivity is maximal, consistent with a reduction in attempt frequency for ion migration when the compound softens due to the increase in lattice parameter. The polyanion chemistry shows less (inverse) correlation with the activation energy as shown in Fig. 6i. Such a divergent evolution suggests that the prefactor can be used to fine-tune the ionic conductivity. Even though the effect of the prefactor can be neglected in the composition region where the magnitude of the ionic conductivity is primarily determined by the activation energy, when the activation energy is reduced to a sufficiently low value (e.g., ~300 meV), the prefactor should be taken into consideration to further enhance the ionic conductivity. While it is generally assumed that changes in the prefactor are related to a softening of the relevant phonon modes that determine the attempt frequency for hopping, they may also reflect variations in effective carrier concentration caused by more or less local Na ordering.

In summary, through experiments, text-mined literature data, and ab-initio modeling, we have quantified the relationship between the various composition variables of NASICON compounds and the Na-ion conductivity. We find that Na-ion conductivity can be optimized by: (1) an average cation radius of slightly above 0.72 Å for NASICONs with a $(SiO_4)_2PO_4$ polyanion composition; (2) high $SiO_4^{4-}$ polyanion content, and (3) a carefully tuned Na content around $Na_3$ per formula unit but depending on the polyanion composition. In addition, by constructing a first-principles synthesis map for the possible chemical space of NASICON compounds, eight NASICONs were successfully synthesized and investigated as solid-state conductors, and room-temperature (~25 °C) ionic conductivity of 1.2 mS cm$^{-1}$ was achieved in $Na_{3.4}Hf_{0.6}Sc_{0.4}ZrSi_2PO_{12}$.

## Methods
### Experiments
For the solid-state synthesis of the NASICON compounds, typical metal oxides or hydroxides ($HfO_2$ (Aldrich, 99.8%), MgO (Aldrich, ≥99.99%), $Sc_2O_3$ (Sigma, 99.9%), $Zr(OH)_4$ (Aldrich, 97%), $SnO_2$ (Alfa Aesar, 99.9%), CaO (Sigma-Aldrich, 99.9%), $In_2O_3$ (Sigma, 99.998%), $Y_2O_3$ (Sigma, nanopowder, <50 nm particle size)) were used as precursors to introduce metal cations. $SiO_2$ (Sigma-Aldrich, nanopowder) and $NaH_2PO_4$ (Sigma, >99%) were used as silicate and phosphate sources. $Na_2CO_3$ (Sigma-Aldrich, >99%) was used as an extra sodium source. In addition, 10% excess $NaH_2PO_4$ was introduced to compensate for the possible sodium and phosphate loss during the high-temperature treatment. The powder mixtures were wet ball-milled (3 mL ethanol per 50 mL jar) with stainless steel balls and jar for 12 h using a planetary ball mill (PM200, Retsch) at 250 rpm for thorough mixing before pressing into pellets. The pelletized samples were first annealed to form the target phase, then grounded with a mortar and pestle, and wet ball-milled (3 mL ethanol per 50 mL jar) again using a high-energy ball mill (SPEX 8000 M Mixer/Mill) with zirconia balls and jar to reduce the particle size. The powder was then dried and pressed into pellets with a uniaxial press. The pellets were again pressed using a cold isostatic press (YLJ-CIP-20B, MTI), then wrapped with Pt foil for the second high-temperature annealing in an alumina crucible. The detailed annealing conditions for the first and second steps are provided in Supplementary Table 10. Synchrotron powder diffraction data of $Na_3HfZr(SiO_4)_2(PO_4)$, $Na_3HfSc(SiO_4)(PO_4)_2$, $Na_{3.4}Hf_{0.6}Sc_{0.4}ZrSi_2PO_{12}$ were collected at beamline 7-BM, National Synchrotron Light Source II, Brookhaven Nation Lab using an average wavelength of 0.6199 Å. Other obtained materials were analyzed using Rigaku Miniflex 600, Bruker D8 Diffractometer with Cu Kα radiation, scanned over 10–70° two theta range, with data points collected every 0.02° 2θ and scan speed of ~0.005° s$^{-1}$. Data analysis was carried out via Rietveld refinement[64] using Fullprof program[65]. Certain steps were carried out

during the refinements with regard to the site occupancies: (1) First the atomic occupancies were set to their nominal compositions. (2) The occupancy of M1, M2 were refined with the constraint that their occupancies sum to 2; (3) The occupancy of Na1, Na2 (and Na3 for monoclinic NASICON) were then refined freely; (4) To ensure the charge neutrality, with the refined occupancy of Na1, Na2, M1 and M2, the Si, P occupancies were manually calculated with the constraint of their occupancies summing to 3; (5) We applied the manually calculated Si, P occupancies to the refined model, and performed step (2) and step (3) again. During this step, the change of Na1, Na2, M1 and M2 occupancies were <1%, therefore charge neutrality was maintained. For pure phosphate compounds, the P occupancy was fixed at 3, and the occupancy of Na1, Na2, M1, M2 were refined with constraints that charge neutrality was ensured, i.e., for $Na_3ScYP_3O_{12}$, $Sc_{Occ}$ + $Y_{Occ}$ = 2, $Na1_{Occ}$ + $Na2_{Occ}$ = 3; $Na_3ScInP_3O_{12}$, $Sc_{Occ}$ + $In_{Occ}$ = 2, $Na1_{Occ}$ + $Na2_{Occ}$ = 3; for $Na_3HfMgP_3O_{12}$, $Hf_{Occ}$ + $Mg_{Occ}$ = 2, $\Delta Hf_{Occ}$ + $\Delta Na1_{Occ}$ + $\Delta Na2_{Occ}$ = 0.

The ionic conductivity was evaluated using EIS at temperatures ranging from -0 to -100 °C. EIS analysis was performed using a Bio-Logic VMP-300 system at the initial open-circuit voltage in the frequency range from 7 MHz to 100 mHz with the application of a 10 mV signal amplitude. The measurements were performed using a BioLogic controlled environment sample holder assembled and sealed in an Ar-filled glovebox. Indium foils were directly pressed onto both sides of the sample pellet as blocking electrodes. Sample pellets are ~ 6 mm in diameter and ~ 1 mm in thickness. For the Sc-doped samples, Na foils were pressed onto both sides of the pellet in a Ar-filled glove box. The Na|NASICON|Na symmetric cell was placed in a customized solid-state pressure cell with an internal pressure of 3 MPa. EIS and cell cycling tests were performed in sequence using the same cell.

The bulk and total ionic conductivities were obtained by fitting the Nyquist plot with two theoretical circuit models: (1) 1 R (bulk) + 1RC (grain boundary) + C for $Na_3HfZrSi_2PO_{12}$ and $Na_3HfScSiP_2O_{12}$ and (2) 2RC (bulk, grain boundary) + C for the others (R: resistance, C: constant phase element, RC: resistance and constant phase element in parallel.) For the Sc-doped samples, only the total resistance (ionic conductivity) was estimated by taking the low-frequency intercept of the semi-circle on the x-axis.

### First-principles calculations

First-principles total energies calculations were performed using the Vienna ab initio simulation package (VASP) with a plane-wave basis set[66]. Projector augmented-wave potentials[67] with a kinetic energy cutoff of 520 eV and the exchange-correlation form in the Perdew–Burke–Ernzerhof generalized gradient approximation (GGA-PBE)[67] were employed for all the structural optimizations and total energy calculations. For all the calculations, a reciprocal space discretization of 25 k-points per Å$^{-1}$ was applied, and the convergence criteria were set to $10^{-6}$ eV for electronic iterations and 0.02 eV/Å for ionic iterations. A rhombohedral conventional cell was used for each of the NASICON structures. The Na-vacancy ordering, cation ordering, and anion ordering were set as the one with the lowest electrostatic energy[68–72]. It should be noted that NASICONs can also exist with other space groups, such as the monoclinic form (C2/c). The monoclinic NASICONs can be regarded as an ordered version of rhombohedral NASICONs[73]. Therefore, the energy difference between rhombohedral and monoclinic NASICONs is expected to be low as it relates to the Na disordering energy. To provide an estimate, the disordering energy for Na is typically 30–40 meV, while the contribution to $E_{hull}$ must be normalized by the number of atoms per formula unit, i.e., ~30–40 meV/(17–21 atoms per formula unit). This energy variation is much smaller than that caused by chemistry variation. With such considerations, all our calculations and analysis were only performed for rhombohedral NASICONs. In addition to the 3881 NASICON structures, all the competing phases in the relevant chemical spaces that are given in The Materials Project[74] were also calculated to construct the phase diagrams. To estimate the synthetic accessibility of the NASICON at finite temperature, we included the ideal configurational entropy and assumed that a NASICON is synthetically accessible when $E_{hull} - TS_{Ideal} \leq 0$. The ideal entropy was calculated by assuming a fully disordered distribution of Na, cation, and anion sites. For a NASICON with the chemical formula $Na_xM_yM'_{2-y}(AO_4)_z(BO_4)_{3-z}$, the ideal entropy is calculated as follows with the units of eV/atom:

$$S_{ideal} = -k_b \frac{1}{x+17} \left( \frac{1}{4} \left( \frac{x}{4} \ln\left(\frac{x}{4}\right) + \left(1 - \frac{x}{4}\right) \ln\left(1 - \frac{x}{4}\right) \right) \right.$$
$$\left. + \frac{1}{2} \left( \frac{y}{2} \ln\left(\frac{y}{2}\right) + \left(1 - \frac{y}{2}\right) \ln\left(1 - \frac{y}{2}\right) \right) + \frac{1}{3} \left( \frac{z}{3} \ln\left(\frac{z}{3}\right) + \left(1 - \frac{z}{3}\right) \ln\left(1 - \frac{z}{3}\right) \right) \right) \quad (1)$$

While this is likely an overestimation of the entropic stabilization that can be achieved, this more generous filter allows us to capture NASICONs which would be synthesizable as metastable phases.

Ab initio molecular dynamic (AIMD), initialized from the DFT relaxed electrostatic ground states of each NASICON composition, was used to investigate the ionic conductivity. All the AIMD calculations were performed in an NVT ensemble with a time step of 2 fs using a Nosé–Hoover thermostat[75] for a period of 160 ps. A minimal Γ-point-only k-point grid was used for all the calculations. The AIMD simulations were run at 600 K, 800 K, 1000 K, 1200 K, and 1500 K. All the data were fitted assuming Arrhenius behavior to obtain the activation energy, diffusion prefactor, and room-temperature (~25 °C) diffusivity.

### Text mining for NASICON conductivity data

In order to gather NASICON conductivity data spanning a variety of chemical systems from scientific literature, we employed a hybrid strategy combining named entity recognition methods along with human assistance. Initially, NASICON-related papers were identified by applying named entity recognition algorithms[43,44,76] and specific selection criteria to over two million material science papers, including (1) any mention of "NASICON" or known NASICON compositions within abstracts and experimental sections and (2) any mention of "conductivity" or properties associated with conductivity in the abstract. In each NASICON-related paper, the NASICON conductivity data were manually curated from tables and figures when available, accounting for the ionic conductivity σ and activation energy $E_a$ corresponding to different temperatures. When high-temperature conductivity and activation energy rather than room-temperature conductivity were reported, the Arrhenius relationship

$$\sigma T = A \exp\left(-E_a / kT\right) \quad (2)$$

was used to project the high temperature conductivity to 300 K, where $A$ is the pre-exponential factor and $k$ is the Boltzmann constant. The compiled NASICON conductivity data have been included in the Source Data file.

## Data availability

Source data are provided with this paper. Other relevant data are available from the corresponding authors upon reasonable request. Source data are provided with this paper.

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

## Acknowledgements

This work was supported by the Samsung Advanced Institute of Technology. The computational analysis was performed using computational resources sponsored by the Department of Energy's Office of Energy Efficiency and Renewable Energy at the National Renewable Energy Laboratory. Computational resources were also provided by the Extreme Science and Engineering Discovery Environment (XSEDE), which is supported by the National Science Foundation grant number ACI1053575 and the National Energy Research Scientific Computing Center (NERSC), a DOE Office of Science User Facility supported by the Office of Science and the U.S. Department of Energy under contract no. DE-AC02- 05CH11231. This research used beamline 7-BM of the National Synchrotron Light Source II, a U.S. Department of Energy (DOE) Office of Science User Facility operated for the DOE Office of Science by Brookhaven National Laboratory under Contract No. DE-SC0012704.

## Author contributions

J.W., B.O., and G.C. initiated and designed the project. G.C. supervised all aspects of the research. B.O. performed the high-throughput DFT calculations and AIMD simulations. B.O. and J.W. analyzed the calculation data. J.W. synthesized the materials and performed characterizations and electrochemical tests with the help of Z.C. and X.Y. X. Y. acquired the synchrotron diffraction data. In addition, T. H. and J.W. performed the text mining and analyzed the data. Y.W., V.L., and H.K. contributed valuable discussion and insights. J.W., B.O., and G.C. wrote the manuscript, and the manuscript was revised by all the authors.

## Competing interests

The authors declare no competing interests.
