## [Peer Review File · Nature Communications]

nature portfolio

Peer Review FileReviewer comments, first round

Reviewer #1 (Remarks to the Author):

The authors of this work report the exploration of the NASICON solid electrolytes compositional map and ionic conduction properties by means of high-throughput DFT and molecular dynamics, supported by experimental report of 8 new compositions. The work is in overall well executed and of interest for the broad battery community of Nature Communications. There are, however, several points that need to be addressed to warrant publication.

- 1) First point, and most critical - the authors claim 8 new NASICON compositions reported in this work. Part of the authors of this work, however, have already reported some of these compositions (5 of the 8) in a previous publication (Ouyang, B., Wang, J., He, T. et al. Synthetic accessibility and stability rules of NASICONs. *Nat Commun* 12, 5752 (2021). <https://doi.org/10.1038/s41467-021-26006-3>). This needs to be clearly pointed out and the manuscript claims to be updated accordingly.
- 2) The authors need to indicate temperature when reporting conductivities (abstract, introduction and several parts in the results).
- 3) In line 73-74 the authors claim that roles of polyanion composition and Na content determining ionic conductivity has not been decoupled. I refer the authors to check recent article where this correlation is studied: Deng, Z., Mishra, T.P., Mahayoni, E. et al. Fundamental investigations on the sodium-ion transport properties of mixed polyanion solid-state battery electrolytes. *Nat Commun* 13, 4470 (2022). <https://doi.org/10.1038/s41467-022-32190-7>
- 4) Tables of refined atom occupancies for new compositions. Considering there are secondary phases present, can the authors be certain of the reported stoichiometries? Authors may consider using synchrotron PXRD or Neutron Powder Diffraction, to certainly assess reported stoichiometries, since isoelectronic elements are present in the compounds, and laboratory PXRD may be not sufficient to ascertain their independent fraction occupancies.
- 5) Authors should include in their study, the influence of the different compositions in the Na occupancies in their different positions (at least in the monoclinic crystal structures), which will also impact their conduction properties.
- 6) Errors should be added to the calculated activation energies and conductivities, so the drafted conclusions from these data are valid.
- 7) I encourage the authors to use element-sensitive probes to study the conductivity properties of their materials (solid-state NMR, for example), specially when impurities are present that could alter their properties.
- 8) The authors should state the temperature at which Na plating/stripping measurement is carried out, and the dimensions of the pellets employed for electrochemical measurements.
- 9) I recommend the authors to follow common / good practice when reporting Nyquist plots, to use proportional axis, so the shape of the data can be compared and analysed adequately.
- 10) Figure S6 plot is empty on the supplied document.

Dr. Marco Amores, Eurecat - Technology Centre of Catalonia

Reviewer #2 (Remarks to the Author):

This manuscript has used large-scale first-principle calculations in conjunction with experimental synthesis and characterization to provide a systematic understanding of Na-ion conductivity on NASICON compounds.

My report on this manuscript is as follows:

1. I want to highlight an interesting similar published work by the same authors in nature communications previously. Ouyang et al., Synthetic accessibility and stability rules of NASICONs, Nat. Comm., (2021) 12:5752. (<https://www.nature.com/articles/s41467-021-26006-3#Sec12>) From my understanding, the previous work claimed to synthesize Na₃HfZrSi₂PO₁₂ and Na₃HfScSi₂PO₁₂ successfully and showed the XRD results in figure 3(c). In this study, it seems that the authors find these compounds again. This needs to be mentioned that these compounds were found earlier.
2. The authors claim to discover in the current work that the optimal Na content is approximately $x = 3$ but depends on the polyanion chemistry. This I find to be an overlapping claim with the previously published work by Ouyang et al. (<https://www.nature.com/articles/s41467-021-26006-3#Sec12>). In the previous work, they even have the text mining results Fig 3(a) to show the conductivity peaks around Na \sim 3. This also has been shown previously in older literatures (<https://www.sciencedirect.com/science/article/pii/S0022459688900655>).
3. Can the authors provide their rationale for choosing these 16 cations? Specifically, why were divalent cations such as Co²⁺ and Ni²⁺ and trivalent cations such as Al³⁺ not considered?
4. The authors mention that Na_{3.4}Hf_{0.6}Sc_{0.4}ZrSi₂PO₁₂ exhibits one of the highest conductivities of 1.2 mS Cm⁻¹. Is this conductivity the highest conductivity among all NASICONs at room temperature? Since temperature is an important factor on the conductivity of the NASICONs it is critical that the measurement temperatures and the claims should include the parameters. Jolley et al. have previously found Co-doped NASICONs with the greatest total conductivity of 1.55 mS Cm⁻¹ (<https://doi.org/10.1007/s11581-015-1498-8>). Furthermore, a conductivity of 2.4 mS/Cm was also found by Zhang et al. (<https://doi.org/10.1002/aenm.201902373>). Therefore, I am doubtful that the claim 1.2 mS Cm⁻¹ is one of the highest conductivities reported in the NASICONs (as mentioned on page 14).
5. The finding that high conductivity is found in compositions with a large cation size until the size reaches an optimal value ($\approx 0.72 \text{ \AA}$) has also been found before by Jolley et al. (<https://doi.org/10.1007/s11581-015-1498-8>). However, in the study by Jolley et al. they considered the combinations with the general formula Na₃ZrMSi₂PO₁₂. In the current study, the authors have studied a large combination of the NASICON structures, thereby giving rise to the general formula Na_xM_yM'₂-y(AO₄)_z(BO₄)_{3-z}. Interesting to note that similar observations have been found. Some of the existing literature should be acknowledged and how this study is different needs to be addressed.
6. The authors claim to discover high silicate content enhances ionic conductivity on page 5. This trend is known and also acknowledged by the authors on page 9 by ref. 41. However, I feel the uniqueness of the current study is that the trend of increase in conductivity with higher silicate content is true irrespective of the cations. Since ref. 41 only considered Zr cations. This needs to be highlighted and would probably make this work more impactful.
7. In the abstract, the authors have mentioned a very strong statement, "Surprisingly, the Na content enhances the conductivity mostly through its effect on the activation barrier, rather than through the carrier concentration." However, the authors haven't proved it. From the AIMD analysis in the main text, they have mellowed this claim significantly by this statement (pg 16), "This finding indicates that the Na content not only determines the charge-carrier concentration, as expected but also affects the conductivity through the migration barrier." Unless I am missing something, the proof of this statement is not solid and needs further clarification.
8. For the purpose of reproduction in the analysis of the Nyquist plots, it would be great if the authors could provide the circuit diagrams. This can be placed in the supplementary information.
9. In Figure S3 of the Nyquist plot, many plots do not show the two semicircles corresponding to the bulk and grain boundary resistivity (ex. Na₃HfZrSi₂PO₁₂). How did the authors deconvolute the contribution of bulk and grain boundary? As such, the same model shouldn't be fitted since the results can vary significantly with the fitting. Did the authors go to a lower temperature to deconvolute the contributions? In the similar light the range of measurement temperature should

be mentioned in the methods section—furthermore, the temperature at which the Nyquist plots of Figure S3 also needs to be mentioned.

10. Methods for the compositional optimization based on text-mined ionic conductivity data are completely missing from the methods section. I think it is important to mention how the text mining was setup.

11. Some minor typographical errors:

a. In page 5. AIMD is referred as Ab-initio Molecular Dynamic instead of Ab-initio Molecular Dynamics.

b. Ref. 41 and 45 are the same.

c. Page 17, ... NASICONs with higher content of SiO_4^{4-} instead of NASICONs with higher content of SiO_4^{4-} .

The manuscript provides insights into a relatively mature discipline of NASICON Na-ion conductors. However, it needs to address some concerns and the parts which overlap with the existing literature. If these questions and concerns can be answered I would recommend for publication.

Reviewer #3 (Remarks to the Author):

In the paper „The design principles for NASICON super-ionic conductors“ the authors present a systematic study to disentangle the structural parameters of NASICON derivatives to pave the way for the synthesis of new highly ionic conductive NASICON structures. Therefore, DFT calculation are made to identify new possible materials, which are afterwards experimentally synthesized and characterized. The findings are compared to a language-driven text-mined historical data and ab initio molecular dynamics simulations. The paper concludes with three general values for the development of new materials: 1) Na content of ~ 3 , 2) cation radius of 0.72 Å and 3) high silicate content. However, some major points need to be improved before publication:

1) The first part of the paper (DFT + synthesis, line 112-203) is consistent in itself by the evaluation of possible compounds and their characterization. However, it does not fit completely in the line of argument as the synthesized compounds are not particular chosen to show systematic variation in the three convoluted parameters (in contrast to the approach taken in AIMD simulations). The scope of this part is rather the identification and synthesis of “new” NASICON compounds. This is especially true as the conclusion drawn in this part do not support the three key findings of the paper or do not provide novelty:

a. The increasing ionic conductivity with silicate content (line 175-176) has already been published in a more systematic study on $\text{Na}_{1+x}\text{Zr}_2\text{SixP}_{3-x}\text{O}_{12}$ (reference 41)

b. The amount of Na atoms was fixed to 3

c. The optimal r_M value seems to be dependent on the selected polyanion mix rather than a general value. For the $(\text{PO}_4)_3$ derivatives the highest ionic conductivity is achieved with a value of 0.773 Å, and for the $\text{SiO}_4(\text{PO}_4)_2$ with 0.728 Å, respectively.

I would suggest publishing this work in a separate manuscript with a different scope.

2) The synthetic part of the Sc-doped species ($\text{Na}_{3+x}\text{Hf}_{1-x}\text{Sc}_x\text{ZrSi}_2\text{PO}_{12}$, line 204-274) may be a good experimental prove for the key findings of the manuscript (rather than the DFT part). Please add and discuss the Na content (> 3) and r_M values and eventually add them to Figure 4. In general, this might be more fitting after the text-mined data and the AIMD simulations.

3) In Figure 4b) and c) a line for the $\text{Na}_3\text{MM}'\text{SiP}_2\text{O}_{12}$ could be helpful to see the trend, especially as these are the materials considered in the AIMD simulations

4) In the AIMD simulations the variation of the Na-content (Figure 6 a, d and g) should also include $(\text{PO}_4)(\text{SiO}_4)_2$ as these compounds are synthesized in the work and are also taken as parameter for the cation size.

5) The highest value of ionic conductivity Figure 6b is at a r_M value of 0.71 Å, but in the conclusions, the value for optimal conductivity is given with 0.72 Å. Please explain the deviation.

6) Line 370: Please add the crucible and ball material for the wet milling process as well as the time and rpm number to the experimental part.

7) In the supporting information:

- a. Figure S3: Please depict the Nyquist plots with orthonormal scales for better comparability of the semi-circle shape and add the temperature (room temperature ?) at which these have been recorded to the caption
- b. Figure S6 is empty. Please add the data.

Response letter

Reviewer #1 (Remarks to the Author):

The authors of this work report the exploration of the NASICON solid electrolytes compositional map and ionic conduction properties by means of high-throughput DFT and molecular dynamics, supported by experimental report of 8 new compositions. The work is in overall well executed and of interest for the broad battery community of Nature Communications. There are, however, several points that need to be addressed to warrant publication.

1) First point, and most critical - the authors claim 8 new NASICON compositions reported in this work. Part of the authors of this work, however, have already reported some of these compositions (5 of the 8) in a previous publication (Ouyang, B., Wang, J., He, T. et al. Synthetic accessibility and stability rules of NASICONs. Nat Commun 12, 5752 (2021). <https://doi.org/10.1038/s41467-021-26006-3>). This needs to be clearly pointed out and the manuscript claims to be updated accordingly.

We thank the reviewer for pointing this out. We have modified the manuscript to clarify that the major novelty in this work is to present a comprehensive structural and electrochemical study for the new NASICONs.

In the abstract: "...*lead to the successful synthesis and electrochemical investigation of several new NASICONs solid-state conductors.*"

On page 5: "*A subset of the predicted compositions is experimentally explored, leading to the successful synthesis of eight NASICONs. Five of these were introduced in our recent work³⁵ and three are new. In this paper we study the electrochemical performance of these eight recently discovered compounds as solid-state conductors, and present detailed crystal structure and electrochemical impedance.*"

On page 6: "*only 23 compounds with Na₃ stoichiometry were further considered, within which the electrochemical performance of 17 compounds has not yet been reported.*"

We also went through the manuscript carefully and made several minor modifications to make sure that all the wording is appropriate.

2) The authors need to indicate temperature when reporting conductivities (abstract, introduction and several parts in the results).

The temperature of the conductivity measurements has been added throughout the manuscript (abstract, introduction, Figure 3, Figure 5, Supplementary Figure 3 and 9). The EIS measurements were performed at room temperature (~ 25 °C), and the Na stripping and plating test was performed in a 25 °C temperature chamber.

3) In line 73-74 the authors claim that roles of polyanion composition and Na content determining ionic conductivity has not been decoupled. I refer the authors to check recent article where this correlation is studied: Deng, Z., Mishra, T.P., Mahayoni, E. et al. Fundamental investigations on the sodium-ion transport properties of mixed polyanion solid-state battery electrolytes. Nat Commun 13, 4470 (2022). <https://doi.org/10.1038/s41467-022-32190-7>

We thank the reviewer for the suggestion. We have already cited this paper on page 10 (ref 19): “In general, the ionic conductivity increases as the silicate content increases.”¹⁹”

The authors of this paper evaluated the Na hopping events within a local migration unit with different occupancies of Si, P and Na. From these results, they draw the conclusion that increasing Si content *locally* can lower the migration barrier for the nearby ion hopping events. However, considering charge neutrality, the Si content of the material is always coupled with the Na content. Increased conductivity in $\text{Na}_{1+x}\text{Zr}_2\text{Si}_x\text{P}_{3-x}\text{O}_{12}$ series with increased x is related to both the Si and Na content effect.

In our work, we decoupled the role of polyanion composition in both experiments and simulations by explicitly keeping other compositional variable constant, i.e., from $\text{Na}_3\text{HfMg}(\text{PO}_4)_3 \rightarrow \text{Na}_3\text{Hf}_{1.5}\text{Mg}_{0.5}(\text{SiO}_4)(\text{PO}_4)_2$, $\text{Na}_3\text{Zr}_{1.5}\text{Mg}_{0.5}(\text{SiO}_4)(\text{PO}_4)_2 \rightarrow \text{Na}_3\text{HfZr}(\text{SiO}_4)_2(\text{PO}_4)$, the conductivity increases with Si content while Na content and average metal radius are close to constant. A similar trend is also observed in the AIMD simulation in Figure 6c, 6f. As such, our manuscript is complimentary to ref 19.

We have modified the manuscript accordingly to clarify the novelty:

On page 4: “However, the specific roles of the polyanion composition and Na content in determining the ionic conductivity have not been **strictly** decoupled.”¹⁹”

On page 18: “**The results in figure 6f show that by keeping the Na content and cation radius constant, the activation energy can be reduced up to ~200 meV solely by increasing the silicate content from 0 to 2/3, implying that the highest conductivity might be obtained in pure silicate NASICONs or NASICONs with SiO_4^{4-} content above 2/3**”

4) Tables of refined atom occupancies for new compositions. Considering there are secondary phases present, can the authors be certain of the reported stoichiometries? Authors may consider using synchrotron PXRD or Neutron Powder Diffraction, to certainly assess reported stoichiometries, since isoelectronic elements are present in the compounds, and laboratory PXRD may be not sufficient to ascertain their independent fraction occupancies.

We thank the reviewer for this suggestion. We collected the synchrotron XRD (sXRD) data for the three high-ionic-conductivity samples, i.e., $\text{Na}_3\text{HfZrSi}_2\text{PO}_{12}$, $\text{Na}_3\text{HfScSiP}_2\text{O}_{12}$ and $\text{Na}_{3.4}\text{Hf}_{0.6}\text{Sc}_{0.4}\text{ZrSi}_2\text{PO}_{12}$, and Figure 2, Supplementary Figure 8 is updated accordingly using the sXRD data. For other compounds, though they were characterized using lab XRD, we applied a suitable step size ($0.02^\circ 2\theta$) and scan rate ($0.005^\circ \text{s}^{-1}$) for data collection (Methods section) which resulted in a good data quality for refinements according to [DOI: 10.1107/s0021889898009856].

Refined structural parameters including stoichiometries for all compounds are now included in Supplementary Table 2-9, 12.

The X-ray atomic scattering factor is related to the atomic number Z. For our samples, the metal species have quite different atomic number, Z, (e.g., Hf and Zr; Hf and Sc; Hf and Mg; Hf and Ca; Zr and Mg; Sc and In; Sc and Y) and therefore can be differentiated and refined appropriately. One exception is Si and P which are close in Z value. Hence, for mixed polyanion compounds we took several steps during the refinement to make sure the refined compositions were meaningful, as described in the Methods section on page 22:

“Certain steps were carried out during the refinements with regard to the site occupancies: 1) First the atomic occupancies were set to their nominal compositions. 2) The occupancy of M1, M2 were refined with the constraint that their occupancies sum to 2; 3) The occupancy of Na1, Na2 (and Na3 for monoclinic NASICON) were then refined freely; 4) To ensure the charge neutrality, with the refined occupancy of Na1, Na2, M1 and M2, the Si, P occupancies were manually calculated with the constraint of their occupancies summing to 3; 5) We applied the manually calculated Si, P occupancies to the refined model, and performed step 2) and step 3) again. During this step, the change of Na1, Na2, M1 and M2 occupancies were <1%, therefore charge neutrality was maintained.”

In summary, for most compounds, the refined composition agrees reasonably with the nominal composition. For $\text{Na}_3\text{Hf}_{1.5}\text{Ca}_{0.5}\text{SiP}_2\text{O}_{12}$ with an apparent $\text{Ca}_3(\text{PO}_4)_2$ secondary phase, the refined Ca content in the NASICON phase is 0.272 rather than 0.5, therefore the average M radius decreases from 0.783 Å (Hf:Ca=1.5:0.5) to 0.75 Å. However, this does not change the trend of \bar{r}_M in Figure 3. We made a clarification on page 10:

“The refined compositions are listed in Supplementary Table 2 – 9 and agree reasonably with the nominal composition, so that the trend of the average cation radius presented in Figure 3 remains valid. Therefore, nominal compositions are used in the analysis for simplicity. However, Na content lower than 3 are observed for the refined compositions, probably due to the Na loss during the high temperature annealing.”

5) Authors should include in their study, the influence of the different compositions in the Na occupancies in their different positions (at least in the monoclinic crystal structures), which will also impact their conduction properties.

We thank the reviewer for the suggestion. The refined Na occupancies of the as-synthesized materials are provided in Supplementary Table 2 – 9, and Supplementary Figure 7. We also discussed the effect of Na occupancies on page 11:

*“It is worth noting that though the nominal Na content of the as-synthesized compounds are all equal to 3, the ratio between Na1, Na2 site occupancies can be different (Supplementary Table 2 – 9, **Supplementary Figure 7**). The Na1 site occupancy decreases as the silicate content increases¹⁹: the three NASICONs with pure phosphates have Na1 occupancy of ~ 0.8, whereas the monoclinic $\text{Na}_3\text{HfZr}(\text{SiO}_4)_2(\text{PO}_4)$ phase with the highest conductivity exhibits the lowest Na1 occupancy of ~ 0.63.”*

6) Errors should be added to the calculated activation energies and conductivities, so the drafted conclusions from these data are valid.

Error bars have been added for all calculated conductivities and activation energies in Figure 6 according to the method developed by Mo et al. [*npj Computational Materials*, 4, 18 (2018)]. The fitting parameters and errors for all Nyquist plots are also summarized in Supplementary Table 11. We thank the reviewer for the suggestion to make our work more rigorous.

7) I encourage the authors to use element-sensitive probes to study the conductivity properties of their materials (solid-state NMR, for example), specially when impurities are present that could alter their properties.

We thank the reviewer for this suggestion to improve our paper, and we agree that solid-state NMR may provide more insights on the microscopic ion conducting mechanism. In this work we

address materials' macroscopic Na diffusion as solid conducting membranes and we have provided combined EIS and AIMD data to support our arguments. Therefore, due to the limited length of the paper and limited instrument access, we are not able to add another section for NMR in the current work.

8) The authors should state the temperature at which Na plating/stripping measurement is carried out, and the dimensions of the pellets employed for electrochemical measurements.

The Na plating/stripping measurement was carried out at 25 °C in a temperature chamber, and we have updated the figure caption accordingly. The pellets are ~ 6 mm in diameter and ~ 1 mm in thickness, which is added in the Methods section as well (page 23).

9) I recommend the authors to follow common / good practice when reporting Nyquist plots, to use proportional axis, so the shape of the data can be compared and analysed adequately.

All Nyquist plots including Figure 5, Supplementary Figure 3,5 and 9 have been replotted with proportional axes.

10) Figure S6 plot is empty on the supplied document.

We thank the reviewer for pointing out the problem. Figure S6 has now been updated and reordered as Supplementary Figure 9.

Dr. Marco Amores, Eurecat - Technology Centre of Catalonia

Reviewer #2 (Remarks to the Author):

This manuscript has used large-scale first-principle calculations in conjunction with experimental synthesis and characterization to provide a systematic understanding of Na-ion conductivity on NASICON compounds.

My report on this manuscript is as follows:

1. I want to highlight an interesting similar published work by the same authors in nature communications previously. Ouyang et al., Synthetic accessibility and stability rules of NASICONs, Nat. Comm., (2021) 12:5752. (<https://www.nature.com/articles/s41467-021-26006-3#Sec12>) From my understanding, the previous work claimed to synthesize Na₃HfZrSi₂PO₁₂ and Na₃HfScSiP₂O₁₂ successfully and showed the XRD results in figure 3(c). In this study, it seems that the authors find these compounds again. This needs to be mentioned that these compounds were found earlier.

We thank the reviewer for pointing this out. We have modified the manuscript to clarify that the major novelty in this work is to present a comprehensive structural and electrochemical study for the new NASICONs.

In the abstract: "...**lead to the successful synthesis and electrochemical investigation of several new NASICONs solid-state conductors.**"

On page 5: *“A subset of the predicted compositions is experimentally explored, leading to the successful synthesis of eight NASICONs. Five of these were introduced in our recent work³⁵ and three are new. In this paper we study the electrochemical performance of these eight recently discovered compounds as solid-state conductors, and present detailed crystal structure and electrochemical impedance.”*

On page 6: *“only 23 compounds with Na₃ stoichiometry were further considered, within which the electrochemical performance of 17 compounds has not yet been reported.”*

We also went through the manuscript carefully and made several minor modifications to make sure that all the wording is appropriate.

2. The authors claim to discover in the current work that the optimal Na content is approximately $x = 3$ but depends on the polyanion chemistry. This I find to be an overlapping claim with the previously published work by Ouyang et al. (<https://www.nature.com/articles/s41467-021-26006-3#Sec12>). In the previous work, they even have the text mining results Fig 3(a) to show the conductivity peaks around $\text{Na} \sim 3$. This also has been shown previously in older literatures (<https://www.sciencedirect.com/science/article/pii/S0022459688900655>).

We agree with the reviewer that the optimal Na content around 3 has already been shown in previous literature, however, we argue that this work brings new insights on the Na content effect in the following two aspects:

- 1) The above-mentioned literature did not deconvolute the effect of Na content and the effect of other compositional variables on ionic conductivity. For example, in the $\text{Na}_{1+x}\text{Zr}_2\text{P}_3\text{-xSi}_x\text{O}_{12}$ solid solution discussed in [<https://www.sciencedirect.com/science/article/pii/S0022459688900655>], the polyanion species varies with Na content. In contrast, in our work, we evaluated the effect of Na content while keeping both the polyanion species and average metal radius constant. (Figure 6)
- 2) Besides, we also pointed out that the optimal Na content might be different for different polyanion compositions, (Figure 4, 6) which is also not explicitly revealed in previous literature.

We also added a few sentences to acknowledge previous work and clarify the differences of our results on page 13:

“Note that the optimal Na content has already been investigated in previous studies.^{19,23,36,45} However, we argue that the optimal value might be different for different polyanion compositions, and in the following section we investigate the Na content effect in more depth by setting other compositional variables constant in AIMD simulations.”

3. Can the authors provide their rationale for choosing these 16 cations? Specifically, why were divalent cations such as Co^{2+} and Ni^{2+} and trivalent cations such as Al^{3+} not considered?

To screen for potential solid electrolytes, we only considered metal species that are electrochemically inactive, therefore Co and Ni were not included. Al^{3+} was indeed considered (Figure 1) but the Al-containing compounds did not pass the screening criteria.

We thank the reviewer for pointing it out, and the manuscript is updated accordingly on Page 6: “we considered SiO_4^{4-} , PO_4^{3-} and SO_4^{2-} as possible polyanions and 16 possible electrochemical inactive metals for M and M’.”

4. The authors mention that $\text{Na}_3.4\text{Hf}_0.6\text{Sc}_0.4\text{ZrSi}_2\text{PO}_{12}$ exhibits one of the highest conductivities of 1.2 mS cm^{-1} . Is this conductivity the highest conductivity among all NASICONs at room temperature? Since temperature is an important factor on the conductivity of the NASICONs it is critical that the measurement temperatures and the claims should include the parameters. Jolley et al. have previously found Co-doped NASICONs with the greatest total conductivity of 1.55 mS cm^{-1} (<https://doi.org/10.1007/s11581-015-1498-8>). Furthermore, a conductivity of 2.4 mS/cm was also found by Zhang et al. (<https://doi.org/10.1002/aenm.201902373>). Therefore, I am doubtful that the claim 1.2 mS cm^{-1} is one of the highest conductivities reported in the NASICONs (as mentioned on page 14).

The ionic conductivity of 1.2 mS cm^{-1} was measured at room temperature ($\sim 25 \text{ degree C}$), and we have included the temperature information throughout the manuscript.

According to the paper [ACS Energy Lett. 2020, 5, 910–915], conductivity measured from EIS can vary by orders of magnitude depending on the sample preparation procedure, local sample inhomogeneity, employed impedance spectroscopy, cell constant, electrode contact, applied pressure etc. In their study, a large range of 4.5 mS cm^{-1} difference in conductivity and 126 meV difference in activation barrier were found even for an identical sample measured by different groups. Therefore, we suggest that minor difference in conductivity or activation barrier are not that significant without a strict alignment in sample preparation and measuring method, especially when comparing data of different samples from different research groups.

Nonetheless, we did not intend to claim that our sample has the “highest conductivity”. But we think the wording “one of the highest” should be appropriate considering the difference between 1.2 and 1.55 or 2.4 mS cm^{-1} is well within the error range presented in [ACS Energy Lett. 2020, 5, 910–915]. The papers suggested by the reviewer are also acknowledged [ref 46, 47] alongside as references.

5. The finding that high conductivity is found in compositions with a large cation size until the size reaches an optimal value ($\approx 0.72 \text{ \AA}$) has also been found before by Jolley et al. (<https://doi.org/10.1007/s11581-015-1498-8>). However, in the study by Jolley et al. they considered the combinations with the general formula $\text{Na}_3\text{ZrMSi}_2\text{PO}_{12}$. In the current study, the authors have studied a large combination of the NASICON structures, thereby giving rise to the general formula $\text{Na}_x\text{M}_y\text{M}'_{2-y}(\text{AO}_4)_z(\text{BO}_4)_{3-z}$. Interesting to note that similar observations have been found. Some of the existing literature should be acknowledged and how this study is different needs to be addressed.

We thank the reviewer for this suggestion and the manuscript has been updated on Page 18 accordingly: “Our finding is also in line with previous studies indicating an optimal cation radius exists,^{23,47} and here we demonstrate this effect more rigorously by keeping other compositional variables constant.”

6. The authors claim to discover high silicate content enhances ionic conductivity on page 5. This trend is known and also acknowledged by the authors on page 9 by ref. 41. However, I feel the uniqueness of the current study is that the trend of increase in conductivity with higher silicate content is true irrespective of the cations. Since ref. 41 only considered Zr cations. This needs to be highlighted and would probably make this work more impactful.

We thank the reviewer for pointing it out, and we have modified the manuscript accordingly on page 10 and on Page 18:

“In general, the ionic conductivity increases as the silicate content increases,¹⁹ irrespective of the cations.”

“The results in figure 6f show that by keeping the Na content and cation radius constant, the activation energy can be reduced up to ~200 meV solely by increasing the silicate content from 0 to 2/3, implying that the highest conductivity might be obtained in pure silicate NASICONs or NASICONs with SiO_4^{4-} content above 2/3.”

7. In the abstract, the authors have mentioned a very strong statement, “Surprisingly, the Na content enhances the conductivity mostly through its effect on the activation barrier, rather than through the carrier concentration.” However, the authors haven’t proved it. From the AIMD analysis in the main text, they have mellowed this claim significantly by this statement (pg 16), “This finding indicates that the Na content not only determines the charge-carrier concentration, as expected but also affects the conductivity through the migration barrier.” Unless I am missing something, the proof of this statement is not solid and needs further clarification.

In Figure 6a and 6d, conductivities and activation energies change simultaneously with Na content, which proves that Na content can affect the conductivity through the migration energy. Specifically in Figure 6d, when Na content changes from 2 to 2.5, the activation energy decreases about 250 meV (orange triangles), which corresponds to four orders of magnitude increase in conductivity, and that is what we observe in Figure 6a (orange circles). Therefore, it is reasonable to argue that Na content affects the conductivity mostly through its effect on the activation barrier. We also modified the manuscript on page 17 and 18:

“Regarding the Na content effect, in Figure 6a and 6d, conductivities and activation energies vary simultaneously with Na content, indicating that Na content not only determines the charge-carrier concentration, as expected, but also affects the conductivity through the migration barrier. The correlation between Na content and migration barrier can be rationalized as follow: the migration barrier is determined by the size of the bottleneck for ion hopping, which is then controlled by the lattice parameter. For NASICONs with 1Na per formula unit, Na ion sits in the Na1 (6b) site that is face-sharing with two MO_6 , as shown in Figure 1a. Increasing Na content will displace the Na^+ into the Na2 (18e) site, which results in reduced occupancy of the Na1 site and increased electrostatic repulsion between MO_6 units, leading to larger unit cell volume and therefore bottleneck size.³⁷ This trend is supported by the correlation between Na content and lattice parameter, as shown in Supplementary Figure 10. Furthermore, consistent with the text-mined data in Figure 4b, Figure 6a also shows that the optimal Na content for conductivity is polyanion dependent: for phosphates the optimal value is 2.5 while for mixed polyanions the optimal value is near 3. Correspondingly in Figure 6d, the Na content that is optimal for lowering the migration barrier is also lower for phosphates than for mixed polyanions. This result indicates that the optimal Na content for different polyanion composition appears to be mostly driven by Na’s effect on the activation energy rather than by its direct relation to the carrier concentration.”

8. For the purpose of reproduction in the analysis of the Nyquist plots, it would be great if the authors could provide the circuit diagrams. This can be placed in the supplementary information.

We thank the reviewer for the suggestion. The equivalent circuit diagrams used for fitting are now summarized in Supplementary Figure 4, along with fitting parameters summarized in Supplementary Table 11.

9. In Figure S3 of the Nyquist plot, many plots do not show the two semicircles corresponding to the bulk and grain boundary resistivity (ex. $\text{Na}_3\text{HfZrSi}_2\text{PO}_{12}$). How did the authors deconvolute the contribution of bulk and grain boundary? As such, the same model shouldn't be fitted since the results can vary significantly with the fitting. Did the authors go to a lower temperature to deconvolute the contributions? In the similar light the range of measurement temperature should be mentioned in the methods section—furthermore, the temperature at which the Nyquist plots of Figure S3 also needs to be mentioned.

We thank the reviewer for this suggestion. We have added several figures and edited the manuscript to make this part clear:

The Nyquist plots of Supplementary Figure 3 were taken at room temperature ($\sim 25\text{ }^\circ\text{C}$), with the equivalent circuit models provided in Supplementary Figure 4, fitting parameters in Supplementary Table 11.

At room temperature, for samples with high ionic conductivity, e.g., $\text{Na}_3\text{HfZr}(\text{SiO}_4)_2(\text{PO}_4)$ (Supplementary Figure 3), and the Sc-doped ones (Figure 5), the Nyquist plots exhibit a single semicircle, so we used 1R to represent the bulk resistance (high frequency intercept), and 1RC component to represent the grain boundary resistance. For other samples, two semicircles can be observed or partially observed, and the fittings were conducted with 2RC in series to represent bulk and grain boundary contribution, respectively.

However, for $\text{Na}_3\text{HfMg}(\text{PO}_4)_3$ the room temperature Nyquist plot cannot be fitted well because the two semicircles are largely overlapping, therefore we indeed went to a lower temperature ($10\text{ }^\circ\text{C}$) to deconvolute each contribution, (Supplementary Figure 5) and used those fitted parameters as the starting parameters to fit the room temperature Nyquist plot (Supplementary Figure 3) to get a valid result.

According to the “brickwork” model described in [Adv. Mater. 2 (1990) No. 3], the grain boundary capacitance usually lies in the range of 10^{-11} to 10^{-8} F. Our fitting results suggest that the grain boundary capacitance (Q2 in Supplementary Table 11) of all samples are in the range of 10^{-10} to 10^{-9} F, which are well consistent with the literature.

For EIS data used to extract the activation energy, the temperature range is added in the method section, and shown in the horizontal axis in Supplementary Figure 6. For some high-temperature data points of which the bulk and grain boundary contribution cannot be deconvoluted, only total conductivities were taken, as also shown in the figure. We have updated the caption to make this information clear.

10. Methods for the compositional optimization based on text-mined ionic conductivity data are completely missing from the methods section. I think it is important to mention how the text mining was setup.

We have included the setup of the text mining on page 12, and detailed methodology could be found in ref 42–44.

“To generalize our findings, we investigated a text-mined dataset containing the experimentally measured conductivity of 475 reported NASICONs. The corresponding papers were identified by screening over two million materials science articles with criteria for specific materials (i.e., NASICONs) and properties (i.e., conductivity) via chemical named entity recognition.”⁴²⁻⁴⁴

The materials synthesis, densification method for the optimized compounds could be found in Methods section and Supplementary Table 10.

11. Some minor typographical errors:

a. In page 5. AIMD is referred as Ab-initio Molecular Dynamic instead of Ab-initio Molecular Dynamics.

Thanks for pointing out the typo. It has been corrected and carefully checked throughout the manuscript.

b. Ref. 41 and 45 are the same.

Thank you for pointing out the mistake. Ref. 45 is now removed.

c. Page 17, ... NASICONs with higher content of SiO_4^{4-} instead of NASICONs with higher content of SiO_4^4 .

Thank you for pointing out the typo, and the formula has now been corrected.

The manuscript provides insights into a relatively mature discipline of NASICON Na-ion conductors. However, it needs to address some concerns and the parts which overlap with the existing literature. If these questions and concerns can be answered I would recommend for publication.

Reviewer #3 (Remarks to the Author):

In the paper „The design principles for NASICON super-ionic conductors“ the authors present a systematic study to disentangle the structural parameters of NASICON derivatives to pave the way for the synthesis of new highly ionic conductive NASICON structures. Therefore, DFT calculations are made to identify new possible materials, which are afterwards experimentally synthesized and characterized. The findings are compared to a language-driven text-mined historical data and ab initio molecular dynamics simulations. The paper concludes with three general values for the development of new materials: 1) Na content of ~3, 2) cation radius of 0.72 Å and 3) high silicate content. However, some major points need to be improved before publication:

1) The first part of the paper (DFT + synthesis, line 112-203) is consistent in itself by the evaluation of possible compounds and their characterization. However, it does not fit completely in the line of argument as the synthesized compounds are not particularly chosen to show systematic variation in the three convoluted parameters (in contrast to the approach taken in AIMD simulations). The scope of this part is rather the identification and synthesis of “new” NASICON compounds. This is especially true as the conclusion drawn in this part does not support the three key findings of the paper or does not provide novelty:

- a. The increasing ionic conductivity with silicate content (line 175-176) has already been published in a more systematic study on $\text{Na}_{1+x}\text{Zr}_2\text{Si}_x\text{P}_{3-x}\text{O}_{12}$ (reference 41)
 - b. The amount of Na atoms was fixed to 3
 - c. The optimal r_M value seems to be dependent on the selected polyanion mix rather than a general value. For the $(\text{PO}_4)_3$ derivatives the highest ionic conductivity is achieved with a value of 0.773 Å, and for the $\text{SiO}_4(\text{PO}_4)_2$ with 0.728 Å, respectively.
- I would suggest publishing this work in a separate manuscript with a different scope.

We agree with the reviewer that the first part does not directly lead to our design principles for NASICON conductors, but it is an important part to make our story complete and coherent. This is important because the conductivity optimization is completely based on what we have actually synthesized. Particularly, our later materials optimization (Figure 5) was based on the identification of the NASICONs presented in the first part. Moreover, the trends observed in the experimentally measured conductivities (Figure 3) revealed that 1) conductivity varies by a large amount with polyanion composition irrespective of the metal of choice and 2) an optimal r_M value exists and inspired our further investigation. Therefore, we think it is appropriate to put it in the beginning of the paper as the first part of a complete story.

Also, we agree with the reviewer that the optimal r_M value may be polyanion dependent. A value > 0.72 Å is optimal for a $(\text{SiO}_4)_2\text{PO}_4$ polyanion systems, but may not be for others. Therefore, we have modified our manuscript accordingly:

On page 5: *“high conductivity is found in compositions with a large cation size until the size reaches an optimal value (≈ 0.72 Å)”*

On page 13: *“**Figure 4c** shows that the maximum conductivity occurs when \bar{r}_M is slightly above 0.72 Å (though the exact optimal value might be dependent on polyanion species)”*

On page 20: *“We find that Na-ion conductivity can be optimized by: 1) an average cation radius of slightly above 0.72 Å for NASICONs with a $(\text{SiO}_4)_2\text{PO}_4$ polyanion composition;”*

2) The synthetic part of the Sc-doped species ($\text{Na}_{3+x}\text{Hf}_{1-x}\text{Sc}_x\text{ZrSi}_2\text{PO}_{12}$, line 204-274) may be a good experimental prove for the key findings of the manuscript (rather than the DFT part). Please add and discuss the Na content (> 3) and r_M values and eventually add them to Figure 4. In general, this might be more fitting after the text-mined data and the AIMD simulations.

We thank the reviewer for the suggestion. We have included all synthesized materials in Figure 4 for comparison, and we modified the manuscript on page 15 to discuss the reasoning of Na content and r_M upon Sc doping:

“Therefore, we attempted to use Sc^{3+} as a large-size dopant with a lower valence to bring both the average cation radius and Na content toward the optimized values. By replacing 0.2, 0.4 Hf^{4+} per f.u. with Sc^{3+} , the average cation radius can be increased from 0.715 Å to 0.719 and 0.722 Å, and the Na content to 3.2 and 3.4, respectively.”

We put this part before the AIMD simulations because it provides direct evidence for our proposed design principles based on experimental and text mining data. The AIMD part not only theoretically confirms our arguments, but also provides more insights into how those compositional variables influence the conductivity, such as the origin of the Na content effect, the bottleneck size effect, the prefactor effect etc., which is more suitable to wrap up our findings into a comprehensive discussion.

3) In Figure 4b) and c) a line for the $\text{Na}_3\text{MM}'\text{SiP}_2\text{O}_{12}$ could be helpful to see the trend, especially as these are the materials considered in the AIMD simulations

We thank the reviewer for the suggestion. A horizontal line for $\text{Na}_3\text{MM}'\text{SiP}_2\text{O}_{12}$ compounds has been added in Figure 4b and 4c.

4) In the AIMD simulations the variation of the Na-content (Figure 6 a, d and g) should also include $(\text{PO}_4)(\text{SiO}_4)_2$ as these compounds are synthesized in the work and are also taken as parameter for the cation size.

We thank the reviewer for the suggestion. Including $(\text{PO}_4)(\text{SiO}_4)_2$ compounds would indeed complete our study, however, due to the limitation imposed by charge neutrality, the Na content in $\text{Na}_{3+2y}\text{Hf}_{2-y}\text{Mg}_y(\text{PO}_4)(\text{SiO}_4)_2$ series cannot go below 3, therefore $y=0$ might be the only point that has a valid conductivity. In this case we could not derive any conclusion from such few data points. Therefore, only the $(\text{PO}_4)_3$ and $(\text{PO}_4)_2(\text{SiO}_4)$ series were plotted in Figure 6a. The conductivity of $\text{Na}_3\text{Hf}_2(\text{PO}_4)(\text{SiO}_4)_2$ ($y=0$) was presented in Figure 6c.

5) The highest value of ionic conductivity Figure 6b is at a rM value of 0.71 Å, but in the conclusions, the value for optimal conductivity is given with 0.72 Å. Please explain the deviation.

The deviation between AIMD derived optimal rM and proposed optimal rM in the conclusions mainly comes from the deviation between simulation and experiment. With such a consideration, AIMD should be used to demonstrate the trend rather than to provide an accurate prediction of the best composition.

6) Line 370: Please add the crucible and ball material for the wet milling process as well as the time and rpm number to the experimental part.

Thank the reviewer for the suggestion. We have added such information in the experimental part on page 21:

"The powder mixtures were wet ball-milled (3 mL ethanol per 50 mL jar) with stainless steel balls and jar for 12 h using a planetary ball mill (PM200, Retsch) at 250 rpm for thorough mixing before pressing into pellets. The pelletized samples were first annealed to form the target phase, then grounded with a mortar and pestle, and wet ball-milled (3 mL ethanol per 50 mL jar) again using a high-energy ball mill (SPEX 8000M Mixer/Mill) with zirconia balls and jar to reduce the particle size. The powder was then dried and pressed into pellets with a uniaxial press. The pellets were again pressed using a cold isostatic press (YLJ-CIP-20B, MTI), then wrapped with Pt foil for the second high-temperature annealing in an alumina crucible. The detailed annealing conditions for the first and second steps are provided in Supplementary Table 10."

7) In the supporting information:

a. Figure S3: Please depict the Nyquist plots with orthonormal scales for better comparability of the semi-circle shape and add the temperature (room temperature ?) at which these have been recorded to the caption

We thank the reviewer for the suggestion. All Nyquist plots in the manuscript have been updated with orthonormal scales, and temperatures (room temperature) are also added in the caption.

b. Figure S6 is empty. Please add the data.

Figure S6 has been updated and reordered as Supplementary Figure 9.

Reviewer comments, second round

Reviewer #1 (Remarks to the Author):

The authors have successfully addressed the comments raised by this reviewer, including sufficient additional experimental work and analyses. I recommend publication of the revised manuscript in Nature Communications.

Dr Marco Amores, Deakin University

Reviewer #2 (Remarks to the Author):

The authors have made substantial changes to the manuscript in the current version. They have addressed all the concerns I had raised before. They have also clarified why their manuscript is novel and the new insights it provides. The current manuscript is now technically sound and impactful. I have no further comments and the manuscript should be considered for publication in the journal nature communications.